# The UEA Digital Humans entry to the GENEA Challenge 2023

Jonathan Windle
University of East Anglia
United Kingdom

Iain Matthews
University of East Anglia
United Kingdom

Ben Milner
University of East Anglia
United Kingdom

Sarah Taylor
Independent Researcher
United Kingdom

## ABSTRACT

This paper describes our entry to the GENEA (Generation and Evaluation of Non-verbal Behaviour for Embodied Agents) Challenge 2023. This year's challenge focuses on generating gestures in a dyadic setting – predicting a main-agent's motion from the speech of both the main-agent and an interlocutor. We adapt a Transformer-XL architecture for this task by adding a cross-attention module that integrates the interlocutor's speech with that of the main-agent. Our model is conditioned on speech audio (encoded using PASE+), text (encoded using FastText) and a speaker identity label, and is able to generate smooth and speech appropriate gestures for a given identity. We consider the GENEA Challenge user study results and present a discussion of our model strengths and where improvements can be made.

## CCS CONCEPTS

• **Computing methodologies** → **Artificial intelligence**; **Animation**.

## KEYWORDS

Speech-to-gesture, 3D pose prediction, gesture generation, Transformer-XL, Self-Attention, Cross-Attention

**ACM Reference Format:**
Jonathan Windle, Iain Matthews, Ben Milner, and Sarah Taylor. 2023. The UEA Digital Humans entry to the GENEA Challenge 2023. In *INTERNATIONAL CONFERENCE ON MULTIMODAL INTERACTION (ICMI '23), October 9–13, 2023, Paris, France.* ACM, New York, NY, USA, 9 pages. https://doi.org/10.1145/3577190.3616116

## 1 INTRODUCTION

Co-speech gesturing contributes to language production and perception during conversation. Gestures can aid conversation turn-taking and listener feedback while also providing semantic context and may be indicative of emotion and emphasis [4, 9, 16, 22]. Speech-driven gesture generation has predominantly focused on estimating motion for monadic speech input of a main-agent, with no knowledge of interlocutor speech and no concept of interaction. Instead,

this year's GENEA challenge focuses on generating gestures in a dyadic setting – predicting a main-agent's motion from the speech of both the main-agent itself and also the speech of the interlocutor.

We introduce a system to the GENEA Challenge 2023 that uses PASE+ [21] speech embeddings in conjunction with FastText [2] word embeddings and a speaker identity label as input to an adapted Transformer-XL [3] architecture to generate smooth, contextually and temporally coherent motion that can adapt to varying lengths of historic context. Specifically, we extend the Transformer-XL model to provide cross-attention with the interlocutor's speech to impart knowledge of both speakers into the prediction.

Video examples and code can be found in the supplement at github.com/JonathanPWindle/uea-dh-genea23.

## 2 BACKGROUND & PRIOR WORK

Many speech-to-motion deep learning techniques are built upon recurrent models, such as bi-directional Long Short-Term Memory models (LSTMs) [5, 7, 23]. Transformer architectures are growing traction in favour of LSTM models in sequence-based AI, with sequence-based motion prediction models already making use of them [1, 10, 15, 24]. Transformer models do not have a concept of temporal position but can effectively model temporal information often using a sinusoidal position embedding which is added to the input.

Transformers rely on attention mechanisms which inform the network which parts of data to focus on [25]. In self-attention, the mechanism is applied to the input sequence to find which elements within the same sequence may relate to each other and which are key to focus on. Conversely, cross-attention is computed for one input source in relation to a separate input source, calculating which elements from one sequence may relate and be important to focus on in another sequence.

To perform sequence-to-sequence generation using a vanilla transformer as defined in Vaswani et al. [25] a sequence is processed over a sliding window with a one-frame stride. For each window of input, one frame of output is generated. This is computationally expensive and window size is limited by the longest input sequence seen during training. As the sequence length increases, the size of the self-attention mechanism also grows exponentially, leading to memory and computational limitations.

The Transformer-XL architecture [3] differs from the traditional transformer architecture in two key ways: 1) Attention is calculated conditioned on the previous context, and 2) the positional encoding uses a learned relative embedding. The Transformer-XL architecture allows for extended attention beyond a fixed length

by using segment-level recurrence with state reuse allowing the alteration of context length. The Transformer-XL can therefore be trained efficiently on small segment lengths while retaining historical influence through the state reuse. As the historic context length can vary, the Transformer-XL introduces a learned, *relative* positional encoding scheme. Due to its improved ability for modelling sequences, we adapt the Transformer-XL architecture for dyadic gesture generation.

## 3 DATA & PREPROCESSING

Our model makes use of the GENEA challenge data [11] derived from the Talking With Hands dataset [12]. This data includes dyadic conversations between a main-agent and interlocutor and consists of high-quality 30fps mocap data in Biovision Hierarchical (BVH) format, with corresponding speech audio and text transcripts. Our task is to generate the main-agent motion conditioned on both main-agent and interlocutor speech. We process both main-agent and interlocutor speech data the same, using all available modalities; motion, speech, transcription and speaker identity.

### 3.1 Motion

Euler angles are required for test submission and are a convenient representation supported by many available 3D animation pipelines. Despite this, Euler angles are discontinuous and difficult for neural networks to learn [28]. We convert rotations to the 6D rotation representation presented by Zhou et al. [28] for their suitability to deep learning tasks. Global skeleton position is also encoded using three $x, y, z$ values. All values are standardised by subtracting the mean and dividing by the variance computed from the training data.

Each identity in the dataset has a skeleton with different bone lengths. Additionally, per-frame joint offsets are also present in the data, possibly to account for bone-stretching in the data capture. Our analysis of these joint offset values revealed very low variance, and setting them to a pre-defined fixed value for all frames did not impact visual performance. We therefore compute one set of bone lengths and offsets per speaker to simplify the training pipeline. We randomly select a sample corresponding to each identity and fix the bone lengths and offsets accordingly using the first data frame. Joint positions can then be computed using the joint angles (measured or predicted) and pre-defined speaker-specific bone measurements.

### 3.2 Speech

*3.2.1 Audio.* We extract audio features using the problem-agnostic speech encoder (PASE+) [21]. PASE+ is a feature embedding learned using a multi-task learning approach to solve 12 regression tasks aimed at encoding important speech characteristics. These 12 tasks include estimating MFCCs, FBANKs and other speech-related information including prosody and speech content.

PASE+ requires audio to be sampled at 16KHz, so we used band-sinc filtering to reduce the audio sample rate from 42KHz to 16KHz. We use the released, pre-trained PASE+ model to extract audio feature embeddings of size 768 that represents a 33ms window of audio to align with the 30 fps motion. The weights for this model are not updated during training.

*3.2.2 Text.* We extract features from the text transcriptions using the FastText word embedding described by Bojanowski et al. [2] using the pre-trained model released by Mikolov et al. [17]. For each spoken word, we extract the word embedding and align the embedding values to each 33ms window of motion. If no word is spoken at a given frame then a vector of zero values is passed. When a word is spoken across multiple frames, the vector is repeated for the appropriate number of frames.

## 4 METHOD

We adapt the Transformer-XL [3] architecture for speech-driven gesture generation. Specifically, we modify this architecture to use both self and cross-attention. The advantage of the Transformer-XL architecture is that it allows us to model the longer term relationship between speech and gesture for input of any duration.

Our feature extraction process, shown in Figure 1, is used to generate a feature vector $\mathbf{X}$ of length $w$ for both the main-agent and interlocutor. These features are then passed to our model as shown in our overview Figure 2 where they are processed using a number os *Self-Attention Blocks* and *Cross-Attention Blocks*.

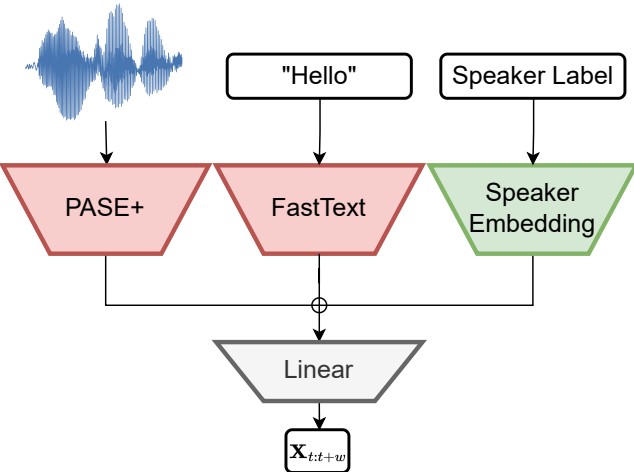

**Figure 1: Outline of our data processing pipeline. Our process takes as input, $w$ frames starting at frame $t$ of speech audio, text transcript and a speaker identity label to generate a feature vector X. We use pre-trained models for the audio and text inputs. Red box defines frozen weights.**

### 4.1 Feature Extraction

We segment the input into non-overlapping segments of length $w$ frames. For each segment, an input feature vector $\mathbf{X}$ is generated and used to predict $\mathbf{Y}$, a sequence of poses of length $w$. Our model is called for each $w$-frame feature vector $\mathbf{X}$. In a speech sequence of length $T$, it is therefore called $\lceil \frac{T}{w} \rceil$ times.

For each segment, we extract audio (PASE+) features $a_{t:t+w}$, and text (FastText) features $f_{t:t+w}$ as described in Section 3.2, where $t$ represents the start frame of a window $w$. For each utterance, there is also a speaker label provided. This is a unique ID which we pass to a learned embedding layer. The embedding layer acts as a lookup

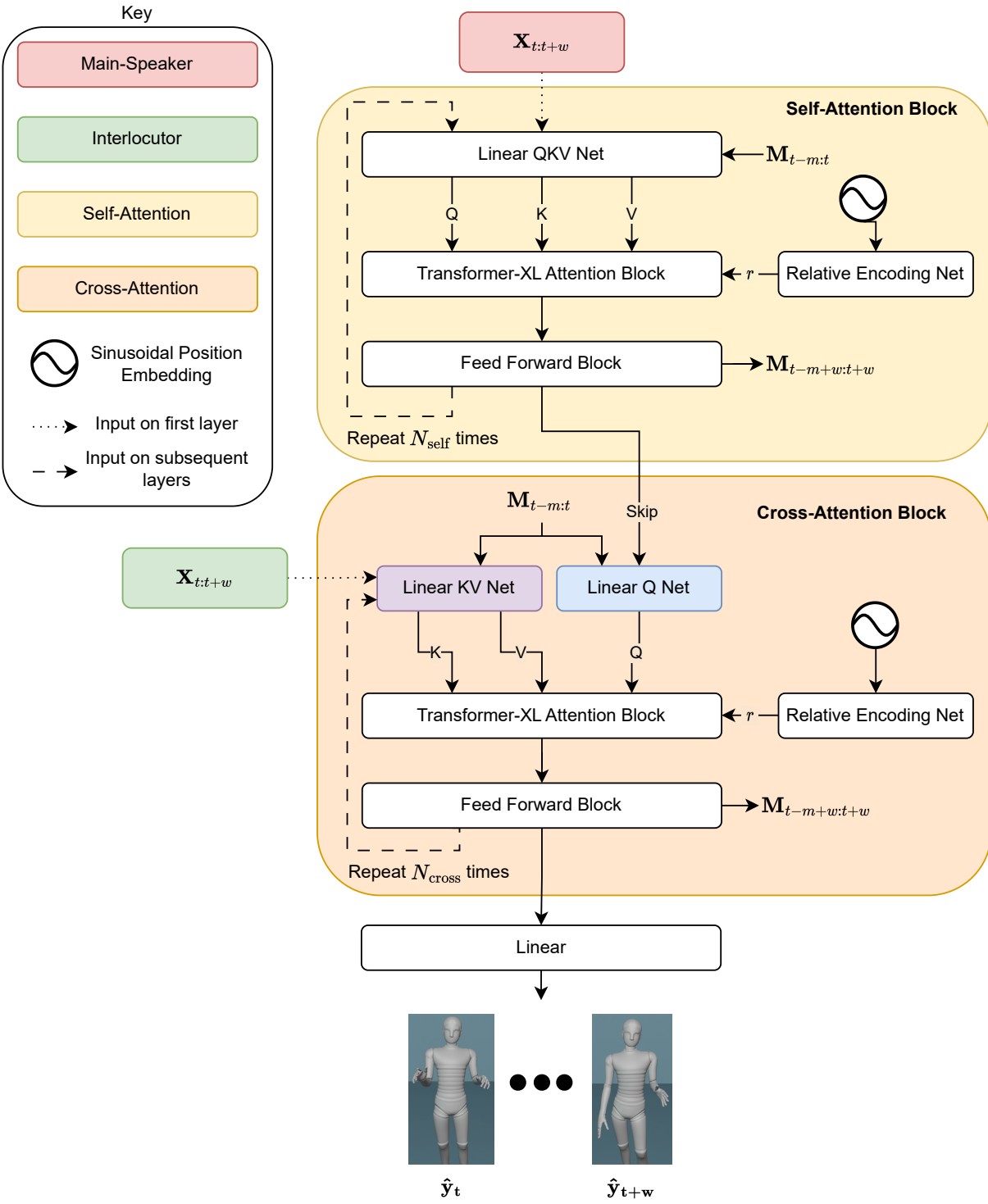

**Figure 2: Outline of our prediction model which takes as input, $w$ motion frames worth of encoded conditioning information starting at time $t$ and predicts $w$ frames of body motion. We show a self-attention block and cross-attention block, where we extract $Q, K, V$ vectors using main-agent or interlocutor speech according to the attention type conditioned on previous $m$ number of hidden states M. These vectors are passed to the Transformer-XL attention block to calculate attention before being fed into a feed-forward block. A final linear layer predicts $w$ poses $\hat{\mathbf{y}}_{\mathbf{t:t+w}}$.**

table for learned feature embeddings that are representative of each speaker style. The trainable weights ensure that two speakers with similar gesture styles are close in the latent embedding space, and conversely, those with different gesturing styles are far apart.

Each modality is extracted and concatenated into a single feature vector $\mathbf{X}$ as shown in Figure 1. Feature vectors for both the main-agent and the interlocutor are extracted in the same way using the same learned weights. This is because a speaker may appear as the main-agent in some sequences and the interlocutor in others.

## 4.2 Self-Attention

As shown in Figure 2, we process the features from the main-agent using a self-attention block. The attention score is defined in Vaswani et al. [25] as:

$$\text{Attention}(Q, K, V) = \text{softmax}(\frac{QK^T}{\sqrt{d_k}})V$$

Where Query $Q$, Key $K$, and Value $V$ are all vectors and queries and keys are of dimension $d_k$, and values of dimension $d_v$. These vectors are often linear projections of an input vector into their respective dimensions $d$.

When calculating attention scores in the Transformer-XL model, historic context is included using segment-level recurrence with state reuse. This is achieved by caching previous hidden state sequences which can be used when processing future segments. When no historic context is present at the start of the speech sequence, our Transformer-XL extracts $Q, K$ and $V$ vectors from the *main-agent* inputs alone. The historic context from processed segments $\mathbf{M}$ of length $m$ is cached as each segment is processed. $Q, K$ and $V$ vectors are then extracted from the subsequent inputs, conditioned on previous context. This process is completed using a Linear QKV Net shown in Figure 2 which is a single linear layer.

Transformer models do not have inherent knowledge of positional order. To ensure temporal coherency, a positional encoding is often added to the input vectors to inject some position context to the model. As the Transformer-XL architecture can have varying lengths of historic context and is not constrained to a maximum length, a learned relative position encoding $r$ is instead utilised. The learned relative encoding is from a single linear layer and takes a sinusoidal position embedding for the full length of context, that is the sum of both memory length available and the query length. Rather than injecting the temporal information to the input before calculating $Q$, $K$ and $V$, which is the approach used in Vaswani et al. [25], the Transformer-XL inputs this information after these vectors have been extracted at the time of calculating the attention score.

Using $Q$, $K$ and $V$ in conjunction with the relative position encoding $r$, we use the *Transformer-XL attention block* to calculate attention vectors. As Figure 2 shows, these attention vectors are then passed to a Feed Forward Block which comprises of two Linear layers, with a ReLU activation on the first output and dropout applied to both.

Each self-attention block has multiple attention heads, each aiming to extract different attention features and a self-attention block is repeated $N_{\text{self}}$ times, with each layer feeding its output to the next. Memory values $\mathbf{M}$ are persisted on a per-layer basis and therefore hidden states are specific to each self-attention block. The length of this memory $m$ can be altered during training and evaluation.

## 4.3 Cross-Attention

While it is reasonable to assume the main-agent speech is driving the majority of the gestures, the interlocutor can also influence the motion of the agent indicating turn taking and backchannel communication. For example, the main-agent might nod to show agreement or understanding when the interlocutor is speaking. Therefore we aim to derive the main source of information driving the motion from the main-agent's speech, but also include the interlocutor's speech. We adapt the Transformer-XL to not only compute self-attention over the main-agent inputs, but to also utilise cross-attention from the *interlocutor* while maintaining segment-level recurrence and relative position encoding. This cross-attention block is shown in Figure 2.

Cross-attention is an attention mechanism where the Query $Q$ is extracted from the input source and the Key $K$ and Value $V$ are extracted from an external input element. Our cross-attention block uses a similar approach as the *self-attention block* defined in Section 4.2, but instead has two separate networks to process the inputs; one to extract $Q$ from the *main-agent* self-attention encoding and one to extract $K$ and $V$ derived from the *interlocutor* speech. For each layer of cross-attention blocks, the input to the $Q$ net is a skip connection from the output of the self-attention encoder and therefore remains the same input for all *cross-attention blocks*. The input to the $KV$ net in the first iteration is the interlocutor feature vectors (described in Section 4.1), and the output from a cross-attention block thereafter.

The output from the cross-attention block is then passed to a single linear layer which predicts $\mathbf{Y}$, the standardised 6D rotations of each joint and the global position of the skeleton.

## 4.4 Training Procedure

For each segment of speech of length $w$, we predict the pose represented by a vector of joint rotations $\hat{\mathbf{Y}}$ of length $w$. In motion synthesis it is common to include both geometric and temporal constraints in the loss function to ensure that the model generates output that is both geometrically and dynamically plausible [6, 24, 26]. Our loss function $L_c$ comprises multiple terms including a $L_1$ loss on the rotations ($L_r$), positions ($L_p$), velocity ($L_v$), acceleration ($L_a$) and kinetic energy ($L_{v^2}$) of each joint. If we take $\mathbf{y_r}$ and $\hat{\mathbf{y}}_\mathbf{r}$ to be natural mocap and predicted 6D rotations respectively; $\mathbf{y_p}$ and $\hat{\mathbf{y}}_\mathbf{p}$ to to be positions in world space computed using forward kinematics given the predicted joint angles and the pre-defined speaker-specific bone lengths, we use the following loss function:

$$
\begin{aligned}
L_r &= L_1(\mathbf{y_r}, \hat{\mathbf{y}}_\mathbf{r}) \\
L_p &= L_1(\mathbf{y_p}, \hat{\mathbf{y}}_\mathbf{p}) \\
L_v &= L_1(f'(\mathbf{y_p}), f'(\hat{\mathbf{y}}_\mathbf{p})) \\
L_{v^2} &= L_1(f'(\mathbf{y_p})^2, f'(\hat{\mathbf{y}}_\mathbf{p})^2) \\
L_a &= L_1(f''(\mathbf{y_p}), f''(\hat{\mathbf{y}}_\mathbf{p})) \\
L_c &= \lambda_p L_p + \lambda_v L_v + \lambda_a L_a + \lambda_r L_r + \lambda_{v^2} L_{v^2}
\end{aligned}
\tag{1}
$$

Where $f'$ and $f''$ are the first and second derivatives respectively. Each term has a $\lambda$ weighting to control the importance of each term in the loss.

Table 1 summarises the parameters used, optimised using a random grid search parameter sweep. These settings were chosen using a combination of low validation loss values and quality of the predicted validation sequences as observed by our team. We train our model for 1770 epochs using the AdamW [14] optimiser and found that a segment length $w$ of 90 frames and memory length $m$ of 180 frames was optimal. The *Feed Forward Blocks* used in both self and cross-attention layers are comprised using the same topology and size.

| Hyperparameter | | Value |
|---|---|---|
| TransformerXL | Head Dimension | 32 |
| | Number Heads | 32 |
| | Self-Attention Layers ($N_{\text{self}}$) | 4 |
| | Cross-Attention Layers ($N_{\text{cross}}$) | 2 |
| Feed Forward Block | Dropout | 0.2 |
| | Hidden Size | 4096 |
| Embeddings | Feature Embedding | 1024 |
| | Speaker Embedding | 8 |
| Training | Batch Size | 32 |
| | Learning Rate | 0.00001 |
| | $\lambda_r$ | 1 |
| | $\lambda_p$ | 0.01 |
| | $\lambda_v, \lambda_a$ | 0.5 |
| | $\lambda_{v^2}$ | 0.2 |
| Context | Segment Length ($w$) | 90 frames |
| | Memory Length ($m$) | 180 frames |

**Table 1: Training hyperparameters.**

## 5 RESULTS

Our approach is evaluated in conjunction with the GENEA Challenge 2023 [11]. Each challenge participant submitted 70 BVH files for main-agent motion generated using the speech of the main-agent and interlocutor for each interaction. Using these submitted BVH files, motion is rendered on the same character for comparison. There are three studies of interest in this challenge; human likeness, appropriateness to speech and appropriate to interlocutor. Each challenge participant is assigned a unique ID to provide anonymity during the evaluation process, our ID which will be used in Figures and Tables throughout is **SJ**. **NA** denotes natural motion of the mocap sequences, **BD** and **BM** are baseline systems in a dyadic and monadic setting respectively. We give a brief overview of each evaluation method, however, we strongly recommend also reading the main challenge paper [11] for full details.

| Condition | Human-likeness | |
|---|---|---|
| | Median | Mean |
| NA | 71 ∈ [70, 71] | 68.4±1.0 |
| SG | 69 ∈ [67, 70] | 65.6±1.4 |
| SF | 65 ∈ [64, 67] | 63.6±1.3 |
| **SJ** | 51 ∈ [50, 53] | 51.8±1.3 |
| SL | 51 ∈ [50, 51] | 50.6±1.3 |
| SE | 50 ∈ [49, 51] | 50.9±1.3 |
| SH | 46 ∈ [44, 49] | 45.1±1.5 |
| BD | 46 ∈ [43, 47] | 45.3±1.4 |
| SD | 45 ∈ [43, 47] | 44.7±1.3 |
| BM | 43 ∈ [42, 45] | 42.9±1.3 |
| SI | 40 ∈ [39, 43] | 41.4±1.4 |
| SK | 37 ∈ [35, 40] | 40.2±1.5 |
| SA | 30 ∈ [29, 31] | 32.0±1.3 |
| SB | 24 ∈ [23, 27] | 27.4±1.3 |
| SC | 9 ∈ [ 9,  9] | 11.6±0.9 |

**Table 2: Summary statistics of user-study ratings from the human-likeness study, with confidence intervals at the level $\alpha$ = 0.05. Conditions are ordered by decreasing sample median rating. Our model results are highlighted in pink . Table and caption from [11].**

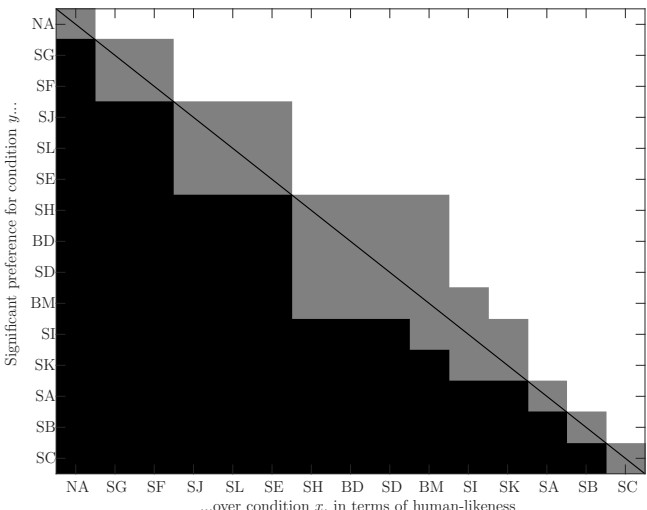

**Figure 3: Significance of pairwise differences between conditions in human-likeness study. White means that the condition listed on the $y$-axis rated significantly above the condition on the $x$-axis, black means the opposite ($y$ rated below $x$), and grey means no statistically significant difference at the level $\alpha$ = 0.05 after Holm-Bonferroni correction. Conditions are listed in the same order as in Table 2. Figure and caption from [11].**

## 5.1 Human Likeness

This user-study aims to evaluate how human-like the motion generated is, independent of the speech. Although each comparison system motion corresponds to the same input speech and conditioning, these sequences were muted to ensure ratings can only depend on the motion seen in the videos. 8 systems were compared at any one time and participants were asked "Please indicate on a sliding scale how human-like the gesture motion appears". Study participants gave their ratings in response to this question on a scale from 0 (worst) to 100 (best).

Summary statistics (median, mean) are shown in Table 2 and significance comparisons are provided in Figure 3. Our system (**SJ**) was evaluated to be the third highest ranking of submitted systems with regards to mean and median human likeness score. Figure 3 shows only **NA**, **SG** and **SF** are significantly better than our system. Our system scores significantly higher than 9 other systems, including both baseline systems.

| Condi- | MAS | Pref. | Raw response count | | | | | |
|---|---|---|---|---|---|---|---|---|
| tion | | matched | 2 | 1 | 0 | −1 | −2 | Sum |
| NA | 0.81±0.06 | 73.6% | 755 | 452 | 185 | 217 | 157 | 1766 |
| SG | 0.39±0.07 | 61.8% | 531 | 486 | 201 | 330 | 259 | 1807 |
| SJ | 0.27±0.06 | 58.4% | 338 | 521 | 391 | 401 | 155 | 1806 |
| BM | 0.20±0.05 | 56.6% | 269 | 559 | 390 | 451 | 139 | 1808 |
| SF | 0.20±0.06 | 55.8% | 397 | 483 | 261 | 421 | 249 | 1811 |
| SK | 0.18±0.06 | 55.6% | 370 | 491 | 283 | 406 | 252 | 1802 |
| SI | 0.16±0.06 | 55.5% | 283 | 547 | 342 | 428 | 202 | 1802 |
| SE | 0.16±0.05 | 54.9% | 221 | 525 | 489 | 453 | 117 | 1805 |
| BD | 0.14±0.06 | 54.8% | 310 | 505 | 357 | 422 | 220 | 1814 |
| SD | 0.14±0.06 | 55.0% | 252 | 561 | 350 | 459 | 175 | 1797 |
| SB | 0.13±0.06 | 55.0% | 320 | 508 | 339 | 386 | 262 | 1815 |
| SA | 0.11±0.06 | 53.6% | 238 | 495 | 438 | 444 | 162 | 1777 |
| SH | 0.09±0.07 | 52.9% | 384 | 438 | 258 | 393 | 325 | 1798 |
| SL | 0.05±0.05 | 51.7% | 200 | 522 | 432 | 491 | 170 | 1815 |
| SC | −0.02±0.04 | 49.1% | 72 | 284 | 1057 | 314 | 76 | 1803 |

**Table 3: Summary statistics of user-study responses from the appropriateness to speech study, with confidence intervals for the mean appropriateness score (MAS) at the level $\alpha = 0.05$. "Pref. matched" identifies how often test-takers preferred matched motion in terms of appropriateness, ignoring ties. Our model results are highlighted in pink. Table and caption from [11].**

## 5.2 Speech Appropriateness

To measure appropriateness of gestures to speech, participants were asked to view two videos and answer "Which character's motion matches the speech better, both in terms of rhythm and intonation and in terms of meaning?". Both video stimuli are from the same condition and thus ensure the same motion quality, but one matches the speech and the other is mismatched, generated from an unrelated speech sequence. Five response options were available, namely "Left is clearly better", "Left is slightly better", "They are equal", "Right is slightly better", and "Right is clearly better". Each answer is assigned a value of -2, -1, 0, 1, 2 where a negative value is given for a preference to mismatched motion and a positive value for a preference to matched motion.

Table 3 provides summary statistics and win rates, Figure 4 visualises the response distribution and Figure 5 shows significance comparisons. Our approach (**SJ**) ranked second in the submitted systems. Figure 5 shows that there are few significant differences between pairwise systems. Only **SG** and the natural mocap (**NA**) rank significantly better than our system. Again, our system ranks significantly better than 9 other conditions including the dyadic baseline system.

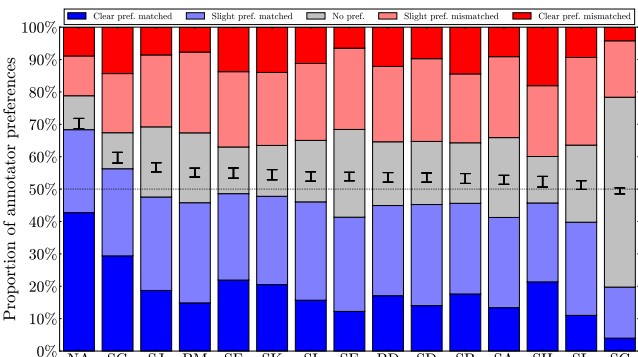

**Figure 4: Bar plots visualising the response distribution in the appropriateness to speech study. The blue bar (bottom) represents responses where subjects preferred the matched motion, the light grey bar (middle) represents tied ("They are equal") responses, and the red bar (top) represents responses preferring mismatched motion, with the height of each bar being proportional to the fraction of responses in each category. Lighter colours correspond to slight preference, and darker colours to clear preference. On top of each bar is also a confidence interval for the mean appropriateness score, scaled to fit the current axes. The dotted black line indicates chance-level performance. Conditions are ordered by mean appropriateness score. Figure and caption from [11].**

## 5.3 Interlocutor Appropriateness

As this year's challenge includes awareness of the interlocutor speech and motion, the appropriateness of the generated main-agent motion to the interlocutor's speech is also evaluated. The was done using a similar technique used for measuring speech appropriateness but differed in several important aspects. The test data contained pairs of interactions, one with matched main-agent and interlocutor interactions and another with the same main-agent speech, but mismatched interlocutor speech. Preference can be quantified for generated motion with matched over mismatched interlocutor behaviour and we can assess how interlocutor behaviour affects the motion.

Our system ranked 8th in this study but only natural mocap, **SA**, **BD** and **SL** are rated significantly higher than it. There is no other significant difference to any other system, except **SH** where we were significantly better. We observe from the statistics in Figure 7 that our system had the lowest number of negative scores (preference for the mismatched dyadic interaction), and a large number of no preference scores.

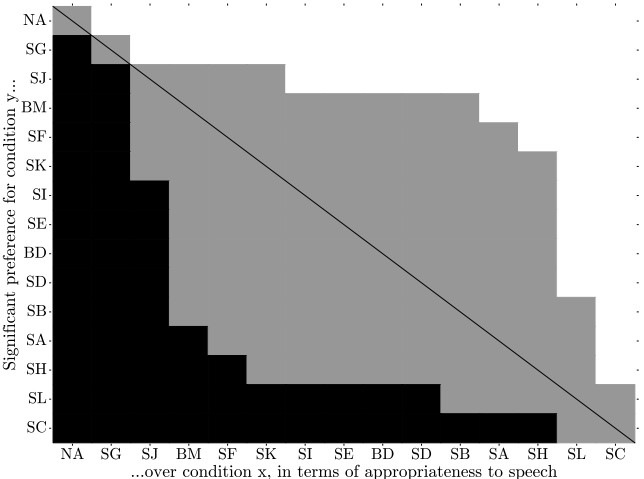

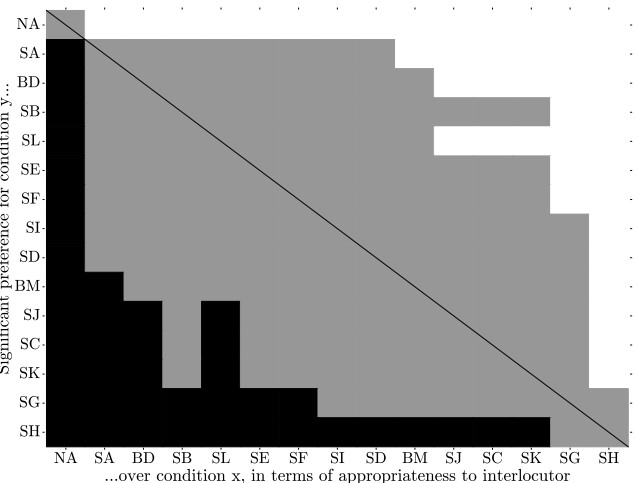

**Figure 5: Significance of pairwise differences between conditions in the appropriateness to speech evaluation. White means that the condition listed on the $y$-axis rated significantly above the condition on the $x$-axis, black means the opposite ($y$ rated below $x$), and grey means no statistically significant difference at the level $\alpha$ = 0.05 after Holm-Bonferroni correction. Conditions are listed in the same order as in Table 3. Figure and caption from [11].**

**Figure 6: Significance of pairwise differences between conditions in the appropriateness to interlocutor study. White means that the condition listed on the $y$-axis rated significantly above the condition on the $x$-axis, black means the opposite ($y$ rated below $x$), and grey means no statistically significant difference at the level $\alpha$ = 0.05 after Holm-Bonferroni correction. Conditions are listed in the same order as in Figure 4. Figure and caption from [11].**

| Cond-ition | MAS | Pref. matched | Raw response count ||||||
| | | | 2 | 1 | 0 | −1 | −2 | Sum |
| --- | --- | --- | --- | --- | --- | --- | --- | --- |
| NA | 0.63±0.08 | 67.9% | 367 | 272 | 98 | 189 | 88 | 1014 |
| SA | 0.09±0.06 | 53.5% | 77 | 243 | 444 | 194 | 55 | 1013 |
| BD | 0.07±0.06 | 53.0% | 74 | 274 | 374 | 229 | 59 | 1010 |
| SB | 0.07±0.08 | 51.8% | 156 | 262 | 206 | 263 | 119 | 1006 |
| SL | 0.07±0.06 | 53.4% | 52 | 267 | 439 | 204 | 47 | 1009 |
| SE | 0.05±0.07 | 51.8% | 89 | 305 | 263 | 284 | 73 | 1014 |
| SF | 0.04±0.06 | 50.9% | 94 | 208 | 419 | 208 | 76 | 1005 |
| SI | 0.04±0.08 | 50.9% | 147 | 269 | 193 | 269 | 129 | 1007 |
| SD | 0.02±0.07 | 52.2% | 85 | 307 | 278 | 241 | 106 | 1017 |
| BM | −0.01±0.06 | 49.9% | 55 | 212 | 470 | 206 | 63 | 1006 |
| SJ | −0.03±0.05 | 49.1% | 31 | 157 | 617 | 168 | 39 | 1012 |
| SC | −0.03±0.05 | 49.1% | 34 | 183 | 541 | 190 | 45 | 993 |
| SK | −0.06±0.09 | 47.4% | 200 | 227 | 111 | 276 | 205 | 1019 |
| SG | −0.09±0.08 | 46.7% | 140 | 252 | 163 | 293 | 167 | 1015 |
| SH | −0.21±0.07 | 44.0% | 55 | 237 | 308 | 270 | 144 | 1014 |

**Table 4: Summary statistics of user-study responses from the appropriateness to interlocutor study, with confidence intervals for the mean appropriateness score (MAS) at the level $\alpha$ = 0.05. "Pref. matched" identifies how often test-takers preferred matched motion in terms of appropriateness, ignoring ties. Our model results are highlighted in pink. Table and caption from [11].**

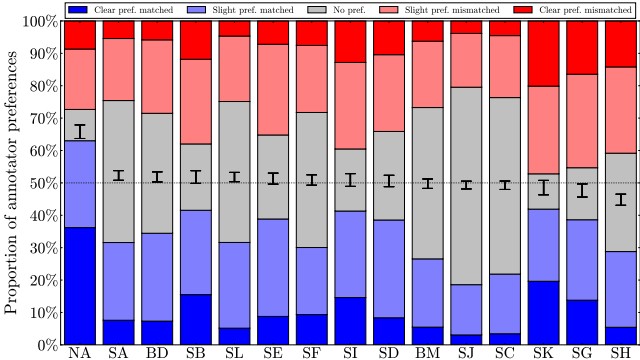

**Figure 7: Bar plots visualising the response distribution in the appropriateness to interlocutor study. The blue bar (bottom) represents responses where subjects preferred the matched motion, the light grey bar (middle) represents tied ("They are equal") responses, and the red bar (top) represents responses preferring mismatched motion, with the height of each bar being proportional to the fraction of responses in each category. Lighter colours correspond to slight preference, and darker colours to clear preference. On top of each bar is also a confidence interval for the mean appropriateness score, scaled to fit the current axes. The dotted black line indicates chance-level performance. Conditions are ordered by mean appropriateness score. Figure and caption from [11].**

## 5.4 Observations

We observe that the animation generated from our model is smooth and temporally coherent without jitter or sudden shifts in motion while maintaining gesture beats in time with speech. Our model appears to reliably and realistically animate beat gestures. Beat gestures are simple and fast movements of the hands and have a close relationship to prosodic activity such as acoustic energy and pitch [20, 27]. The PASE+ model used for encoding audio in our system was trained to estimate prosodic features as one of its downstream tasks, making the derived audio features particularly suitable for animating beat gestures.

We do not expect gestures to occur during every audio beat, but when they happen they should synchronise with the speech. Using the method of motion and audio beat extraction used in the beat align score calculation presented in Liu et al. [13], we can visualise the onset of audio beats and motion gesture over time. Figure 8 shows two well timed gestures for a 3 second audio clip. The utterance of "programs" shows a beat gesture where during the syllable utterance "pro", the speaker moves their right hand from right to left and as the stressed syllable "grams" is spoken, the hand begins to change velocity and move from left to right. We also see an example of muted speech where our model continues to perform well. As there is no speech, there is little to inform gesture, we find the right arm drops to the side, and left arm lowers slightly. However, as the speech begins again, both arms raise in time with the speech.

A difference between natural mocap motion and our generated animation is that the latter does not exhibit sporadic, non-speech related motion such as self-adaptor traits. Self-adaptors are movements that typically include self-touch, such as scratching of the neck, clasping at an elbow, adjusting hair or interlocking fingers [18]. Despite the indirect relationship between these behaviours and speech, these traits are linked to perceived emotional stability of an agent [18] and may influence perceived human-likeness.

## 6 DISCUSSION

Our approach performed well with regards to human-likeness and appropriateness to speech. Our model performed comparably to 10 of the other systems with regards to appropriateness to the interlocutor's speech, but clearly it can be improved in this area. We observe in Figure 7 and Table 4 that, for our system, participants preferred the mismatched stimuli least compared to all other systems (including natural mocap). The majority of responses were tied, meaning that they considered the mismatched stimuli to be of equal appropriateness as the matched animation. It is unclear where this uncertainty stems from and more work is required to evaluate this cause. There may be a lack of influence from the interlocutor speech in this model architecture. There are many ways to incorporate the interlocutor speech in this model, for example including as an extra input to the self-attention rather than as cross-attention or altering skip connections. These ideas or simply increasing the number of cross-attention layers may improve the performance of the appropriateness to the interlocutor.

More experiments are also required to determine the impact of including the interlocutor information on human-likeness and appropriateness to speech as well as appropriateness to interlocutor.

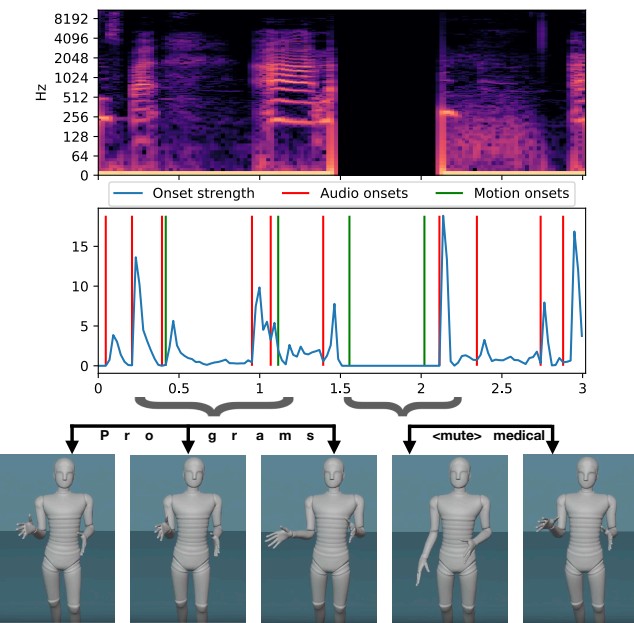

**Figure 8: Generated gestures for given audio beats. Using a 3s audio clip from the test dataset we show the audio spectrogram, as well as aligned audio beat onsets and their corresponding onset strengths as well as motion gesture onset detection of the right wrist using the method of beat detection defined in Liu et al. [13]. We can see during the syllable utterance "pro", the speaker moves their right hand hand from right to left and as the stressed syllable "grams" is spoken, the hand begins to move left to right. When there is silence, the arms begin to rest and again gesture in the next utterance.**

This may have a positive effect on these two evaluations or may limit performance in these areas.

Although our proposed method is deterministic, i.e. the same inputs will always produce the same outputs, it could be possible to incorporate this design into a probabilistic model. For example, this approach could be adjusted to incorporate probabilistic diffusion [8, 19] methods.

## 7 CONCLUSION

We have presented our submission to the GENEA Challenge 2023, a modified Transformer-XL based approach that utilises both self-attention and cross-attention. Our solution generates smooth, temporally coherent animation from the conversational speech of a main-agent and interlocutor. Subjective evaluation results support that our system performs well in regards to human-likeness and appropriateness, ranking third and second respectively when compared to the 14 other systems and baselines and performing significantly better than 9 in both evaluations. Our approach continues to be competitive when evaluating the generated main-agent motion's appropriateness to the interlocutor, where only the natural mocap and 3 systems performed significantly better.

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

Received 20 February 2007; revised 12 March 2009; accepted 5 June 2009