# OpenReview forum: "The UEA Digital Humans entry to the GENEA Challenge 2023"
_ACM.org/ICMI/2023/Workshop/GENEA_Challenge — GENEA Challenge 2023 Mainproceeding_

### Official Review · Reviewer_Kzvu · 2023-08-01
**This paper describes a method to synthesize dyadic gestures given a speech, an audio and a speaker identity label. The authors have used Transformer XL with self attention and cross attention to do so. The paper is well written and technically sound.**

**Rating:** 7
**Confidence:** 3

**Review:**

The paper is well-organized and clearly written. The overall design and technicality of the method seem plausible, and the figures also help to understand the method pretty well. The authors provide sufficient experiments, both statistical and subjective evaluations, in the paper to back up their claims.

A few points to note:
1. I felt that Sections 4.2 and 4.3 could be shortened as the authors only describe the self-attention and cross-attention mechanism in a very generic way which can already be found in other papers.
2. Does using loss on velocity, acceleration, and kinetic energy gets additional benefit? Or is there a chance of overfitting the training data by over-parameterizing these losses?

Overall, I think the solution is well thought out and has potential.

**Nominate For A Reproducibility Award:**

No comment

---

### Official Review · Reviewer_5ajQ · 2023-08-01
**Simple yet promising approach for gesture generation**

**Rating:** 8
**Confidence:** 4

**Review:**

This paper proposes a transformer-based model for gesture generation. The authors improve a Transformer-XL architecture by adding a cross-attention module to incorporate the interlocutor’s information. The technical descriptions are concise and easy to understand. The proposed model is not technically complex but performs well. The experimental results show that the generated gestures are preferably evaluated in terms of both human-likeness and appropriateness.

Comments and questions:
- The motivation for introducing the cross-attention is clearly explained. But it could be more informational if an ablation study of the proposed component.
- In the abstract, it would be better not to spend half of it describing the GENEA challenge itself. Instead, spending much more on describing the original part of the entry would be better.
- To improve the readability, I recommend placing the figures/tables above/below the main texts, not between the texts (Figure 1 is placed between the section title and the main text).

---

### Decision · Program_Chairs · 2023-08-04

**Decision:**

Accept (Main proceeding)

**Comment:**

The paper describes a regression-based system for gesture generation using an extension of the Transformer-XL architecture. All reviewers favoured accepting this paper, awarding good scores. The chairs agree to accept this paper to the Main ICMI Proceedings.

Please read the reviews carefully and use the feedback to revise the paper as relevant for the camera-ready version, for example discussing the effect of the terms in the compound loss. In addition, the authors may wish to consider whether to change the use of the term “ground truth” to another term, since (unlike in many classical machine-learning problems) there is no single, one “true” way to move in gesture generation, even to a given speech. “Natural mocap” is one alternative formulation that can be used instead of writing “ground truth”.